# Ongoing Strategies to Improve Antimicrobial Utilization in Hospitals across the Middle East and North Africa (MENA): Findings and Implications

**DOI:** 10.3390/antibiotics12050827

**Published:** 2023-04-28

**Authors:** Abdul Haseeb, Zikria Saleem, Aseel Fayk Maqadmi, Roaa Abdulrahman Allehyani, Ahmad J. Mahrous, Mahmoud E. Elrggal, Sairah Hafeez Kamran, Manal AlGethamy, Asem Saleh Naji, Abdullmoin AlQarni, Khalid W. Alhariqi, Muhammad Arslan Khan, Kiran Ibrahim, Fahad Raees, Aisha Azmat, Aislinn Cook, Stephen M. Campbell, Giulia Lorenzetti, Johanna C. Meyer, Brian Godman, Catrin E. Moore

**Affiliations:** 1Department of Clinical Pharmacy, College of Pharmacy, Umm Al-Qura University, Makkah 24382, Saudi Arabia; amhaseeb@uqu.edu.sa (A.H.); s438001213@st.uqu.edu.sa (A.F.M.); 120012.roaa@bmc.edu.sa (R.A.A.); ajmahrous@uqu.edu.sa (A.J.M.); merggal@uqu.edu.sa (M.E.E.); 2Department of Pharmacy Practice, Faculty of Pharmacy, Bahauddin Zakariya University, Multan 60800, Pakistan; 3Batterjee Medical College, Jeddah 21442, Saudi Arabia; 4Institute of Pharmacy, Lahore College for Women University, Lahore 54000, Pakistan; sairah.hafeez@lcwu.edu.pk; 5Department of Infection Prevention and Control Program, Alnoor Specialist Hospital, Makkah 24241, Saudi Arabia; mmalgethamy@moh.gov.sa; 6Infectious Diseases Department, Alnoor Specialist Hospital, Makkah 24241, Saudi Arabia; asnaji@moh.gov.sa (A.S.N.); abdullmoina@moh.gov.sa (A.A.); kalhariqi@moh.gov.sa (K.W.A.); 7The Indus Hospital, Bedian Road, Lahore 54000, Pakistan; arsalankhan91352@gmail.com; 8Primary and Secondary Healthcare Department, DHQ Hospital Khushab, Khushab 41200, Pakistan; kiran_paracha@yahoo.com; 9Department of Medical Microbiology, Faculty of Medicine, Umm Al-Qura University, Makkah 24382, Saudi Arabia; frahmed@uqu.edu.sa; 10Department of Physiology, Faculty of Medicine, Umm Al-Qura University, Makkah 24382, Saudi Arabia; aakhan@uqu.edu.sa; 11Centre for Neonatal and Paediatric Infection, St. George’s University of London, London SW17 0RE, UK; aicook@sgul.ac.uk (A.C.); glorenze@sgul.ac.uk (G.L.); camoore@sgul.ac.uk (C.E.M.); 12Health Economics Research Centre, Nuffield Department of Population Health, University of Oxford, Oxford OX1 2JD, UK; 13Centre for Epidemiology and Public Health, School of Health Sciences, University of Manchester, Manchester M13 9PL, UK; stephen.campbell@manchester.ac.uk; 14School of Pharmacy, Sefako Makgatho Health Sciences University, Ga-Rankuwa, Pretoria 0208, South Africa; hannelie.meyer@smu.ac.za; 15South African Vaccination and Immunisation Centre, Sefako Makgatho Health Sciences University, Molotlegi Street, Garankuwa, Pretoria 0208, South Africa; 16Strathclyde Institute of Pharmacy and Biomedical Sciences, Strathclyde University, Glasgow G4 0RE, UK; 17Centre of Medical and Bio-Allied Health Sciences Research, Ajman University, Ajman 346, United Arab Emirates

**Keywords:** antimicrobials, antimicrobial stewardship programs, antimicrobial resistance, Middle East, quality indicators, strategies, surgical site infections, utilization patterns

## Abstract

Antimicrobial resistance (AMR) is an increasing global concern, increasing costs, morbidity, and mortality. National action plans (NAPs) to minimize AMR are one of several global and national initiatives to slow down rising AMR rates. NAPs are also helping key stakeholders understand current antimicrobial utilization patterns and resistance rates. The Middle East is no exception, with high AMR rates. Antibiotic point prevalence surveys (PPS) provide a better understanding of existing antimicrobial consumption trends in hospitals and assist with the subsequent implementation of antimicrobial stewardship programs (ASPs). These are important NAP activities. We examined current hospital consumption trends across the Middle East along with documented ASPs. A narrative assessment of 24 PPS studies in the region found that, on average, more than 50% of in-patients received antibiotics, with Jordan having the highest rate of 98.1%. Published studies ranged in size from a single to 18 hospitals. The most prescribed antibiotics were ceftriaxone, metronidazole, and penicillin. In addition, significant postoperative antibiotic prescribing lasting up to five days or longer was common to avoid surgical site infections. These findings have resulted in a variety of suggested short-, medium-, and long-term actions among key stakeholders, including governments and healthcare workers, to improve and sustain future antibiotic prescribing in order to decrease AMR throughout the Middle East.

## 1. Introduction

Antimicrobial resistance (AMR) is a growing priority globally, increasing morbidity, mortality, and costs [1,2,3,4,5]. According to a recent study, up to 4.95 million (95% uncertainty intervals (UI) 3.62–6.57) deaths globally were associated with bacterial AMR in 2019, with 1.27 million (95% UI 0.911–1.71) deaths directly attributable to AMR [1]. This has prompted global, regional, and national initiatives to try and address the situation [6,7,8,9,10,11], including the development of national action plans (NAPs) to reduce AMR, with countries in the Middle East and North Africa (MENA) being no exception [10,11,12,13,14]. These plans acknowledge that the MENA region contains multiple heterogenic countries with different-sized populations and economic differences [15]. A concern is that AMR rates continue to increase in the MENA region, enhanced by high rates of purchasing antibiotics without a prescription and inappropriate prescribing of antibiotics across all sectors [16,17,18,19,20,21,22,23,24,25,26,27,28,29,30,31,32]; this includes inappropriate prescribing in both hospital and ambulatory care in the MENA region [27,33,34,35,36,37,38,39,40]. Current concerns also include the high rate of prescribing of “Watch” and “Reserve” antibiotics among hospital inpatients, potentially increasing AMR, as well as appreciable prescribing of antibiotics in ambulatory care for self-limiting conditions, including acute respiratory tract infections [36,41,42,43,44,45].

Antibiotics from the “Access” list are considered first-line or second-line treatments for a range of infections and should be routinely available across countries [46,47,48]. There should be limited prescribing of antibiotics in the “Watch” group, as these are considered to have greater resistance potential and toxicity, with very limited prescribing of antibiotics in the “Reserve” antibiotics as these are considered antibiotics of last resort [47,49,50]. The target is at least 60% of antibiotics prescribed should be from the “Access” list [49,51]. However, this is not always the case among countries in the MENA Region [52,53,54].

Inappropriate prescribing of antibiotics in hospitals increased during the COVID-19 pandemic, with high utilization rates across countries in the first COVID-19 wave, including those in the MENA region [55], despite limited evidence of bacterial infections or co-infections in these patients [56,57,58,59]. There are also concerns with high rates of extended prophylaxis postoperatively to prevent surgical site infections (SSIs) among LMICs, with the prevention of SSIs with antibiotics a priority across hospitals and countries, given the increasing prevalence of AMR [60,61,62,63]. However, extended prophylaxis and the use of inappropriate “Watch” antibiotics increase adverse reactions, costs, and AMR, with limited or no improvement in patient outcomes [64,65,66,67,68].

We are now seeing a growth in antimicrobial stewardship programs (ASPs) in hospitals across countries and continents to address concerns with inappropriate prescribing [63,69,70]. These include addressing concerns with extended prophylaxis beyond the first day to prevent SSIs and reducing excessive prescribing of antibiotics generally within hospitals [63,69,71,72]. However, low- and middle-income countries (LMICs) have had barriers to successfully implementing ASPs [73], including having the necessary infrastructure, knowledge, and training in place, together with agreed guidelines, antimicrobial susceptibility testing (AST) data, and champions for ASPs [73,74,75,76,77,78,79]. Nevertheless, this is now changing with ASPs successfully being introduced in hospitals across LMICs, including the MENA region, which will improve future antibiotic prescribing, with the numbers growing over time as healthcare workers increasingly understand their importance [63,70,72,80,81,82].

A range of prescribing and quality indicators have been developed across countries to assess the impact of ASPs [63,70,83,84,85]. There are three main types of indicators (Figure 1), with a clear a priori purpose necessary for developing, collecting, and using these across sectors, including countries in MENA, to improve future antimicrobial prescribing. While different indicators have been used to improve antimicrobial prescribing in hospitals across countries [63,86,87,88,89], any proposed indicator must have clarity, be feasible to implement, and must have easy-to-use, consistent, and reliable tools for valid data collection and management [63,90,91,92]. It is recognized that this is a challenge for many MENA countries, particularly among those with a lower income. Countries are at different stages in their availability of data collection tools, and some still have extensive use of paper-based systems for data collection; however, there is an imperative need for NAPs for AMR to drive down AMR rates through improved antibiotic use [93,94,95,96]. Consequently, routine data collection, and the usefulness of collected data, is likely to change in order to reduce unnecessary prescribing [10,11,13]. In addition, there is increasing use of the WHO Essential Medicines List and the *AWaRe* book to improve empiric prescribing, as well as the monitoring of antibiotic prescribing for a range of infections, alongside pertinent quality indicators [48,49,51]. These are based on key prescribing considerations associated with the guidance (Appendix A) [48,49,97,98].

The first step in improving antibiotic utilization, starting in hospitals across the MENA region, is to describe current utilization patterns. This includes current utilization patterns to prevent SSIs as well as prescribing and quality indications used in practice as part of ASPs alongside other programs to improve future prescribing.

Consequently, we sought to comprehensively describe current antimicrobial consumption patterns among hospitals throughout the MENA region using published point-prevalence surveys (PPS). We also sought to document the prescribing and quality indicators currently being applied across hospitals throughout the MENA region to enhance future antibiotic prescribing. Additionally, we wanted to list ASPs that have been successfully implemented among MENA countries to offer future direction. Since there is currently greater knowledge regarding the prescribing of antibiotics in hospitals compared with ambulatory care across the MENA region, we began the search in hospitals.

As a result, this study attempts to describe the current antimicrobial consumption patterns in hospitals throughout the MENA region using published PPS studies to ascertain the extent to which antibiotics are improperly prescribed postoperatively to avoid SSIs. SSIs account for a significant portion of hospital-acquired infections, enhancing morbidity, mortality, and expenditures [65,99,100,101]; consequently, it is crucial that antibiotic prophylaxis is delivered following correct guidelines to reduce the burden of infections due to SSIs. However, as mentioned, continued administration of antibiotics postoperatively increases adverse reactions and AMR, with little to no effect on further lowering the rate of SSIs [65].

Based on published studies, we also identify potential prescribing and quality indicators that could be applied across hospitals throughout the MENA region to enhance future antibiotic prescribing. In addition, we document ASPs that have been successfully introduced across a range of hospitals in the MENA to offer a future direction, building on recent qualitative studies and reviews [80,81,102].

We will not cover possible programs to address the reluctance among patients in MENA, including those in Jordan and Kuwait, to receive COVID-19 vaccines [103,104,105,106]. Programs to limit the unwarranted use of antibiotics in farming and agriculture in MENA are also ongoing, given AMR concerns [107,108,109,110]. Whilst both areas are very important from a One Health perspective to reduce AMR among patients, both are outside the scope of this study.

## 2. Results

There are currently concerns with high antibiotic utilization rates among hospitals across the MENA region, exacerbated by high rates of empiric prescribing without following current guidelines. Alongside this, there are concerns with the extent of extended prophylaxis beyond the postoperative period to prevent SSIs which, as mentioned, increases adverse reactions, AMR, and costs. However, a number of prescribing and quality indicators are being used across hospitals to monitor and improve prescribing, including those that are part of ASPs. Their impact is documented to provide future guidance.

### 2.1. Current Antimicrobial Utilisation Patterns across the MENA Region

Table 1 documents current antimicrobial utilization patterns among 23 PPS studies that have recently been undertaken across the MENA countries. These incorporate PPS studies from as little as one hospital up to 26 hospitals within a country. In addition, Iskander et al. (2016) documented antimicrobial utilization patterns among 27 non-teaching hospitals in Lebanon, based on actual utilization rates, and subsequently compared these to point prevalence data. The authors found that average consumption levels excluding pediatric cases were 72.56 defined daily dose per 100 bed-days (DDD/100BD), with the most commonly used antibiotics being amoxycillin/clavulanic acid (from the “Access” group), ceftriaxone (from the “Watch” group), amoxycillin (“Access”), and cefuroxime (“Watch”) [111].

We have excluded the MENA hospitals that were part of the 27 Western and Central Asian countries enrolled in the Global PPS of Versporten et al. (2018) [86]. We also excluded the findings from the updated Global PPS analysis, which includes hospitals from Iraq, Iran, Jordan, and Saudi Arabia, and measures the extent of “Access”, “Watch”, and “Reserve” antibiotics prescribed [44], as there was insufficient data from these two consolidated publications for their inclusion in Table 1. However, a summary of their results has been included in our analysis.

In total, over 50% of inpatients in the majority of the surveyed hospitals were prescribed antibiotics, which is comparable to the 42.0% recorded among hospitals from Western and Central Asia taking part in the Global PPS study [86]. Hospitals in Jordan recorded the highest antimicrobial usage rates (98.1%), with hospitals in Tunisia having overall the lowest utilization rates of 39.2–49.2% of patients among the surveyed in-patients (Table 1). Studies evaluating antibiotic utilization among in-patients in Iraq also found high utilization rates at 93.7% of those surveyed.

The most commonly prescribed antibiotics among hospitals in the MENA region reporting their findings (Table 1) were the third-generation cephalosporins (“Watch” group) and the penicillins (“Access” group), followed by metronidazole (“Access” group). This compares to the high levels of “Watch” and “Reserve” antibiotics among hospitals in Iran, Iraq, and Jordan taking part in Pauwels et al.’s (2021) study, which compared with those in Saudi Arabia [44].

### 2.2. Current Length of Antibiotic Prescribing Postoperatively to Prevent SSIs

Table 2 outlines the extent of extended antibiotic prophylaxis among hospitals throughout the MENA region. This is similar to the findings among hospitals from West and Central Asia in the consolidated findings of Versporten et al. (2018) [86]. However, this is not always the case, as seen in the study of Alnajjar et al. (2020), where 100% of mothers undergoing a Cesarean section were just administered a single dose of cefazolin within one hour of skin incision to prevent SSIs [62].

Third-generation cephalosporins, including ceftriaxone (“Watch” group), metronidazole (“Access” group), and the penicillins, including co-amoxiclav (usually “Access” group), were typically administered for surgical antibiotic prophylaxis (SAP).

Published justifications for extended prophylaxis included healthcare professionals (HCP) resistance to change, hospital overcrowding, worries about hospital cleanliness, concerns about proper aseptic techniques not being used during operations, physician ignorance of antibiotics, worries about malnutrition in some patients, and patient expectations.

Appendix A discusses some of the concerns regarding SSIs across the MENA region in addition to extended prophylaxis and potential ways forward to improve future antimicrobial prescribing. There is also a need to strengthen patient education regarding the role of preoperative antibiotics in preventing SSIs, given concerns with limited patient knowledge currently in this area [132].

### 2.3. Prescribing and Quality Indicators

A number of prescribing and quality indicators have been used among hospitals across the MENA region to improve future prescribing (Table 3). These reflect increasing activities among hospitals across MENA to improve future prescribing of antimicrobials and reduce AMR.

However, a major concern across a number of countries in MENA is the current lack of electronic healthcare systems to routinely track prescribing practices against agreed indicators. This is likely to change as more applications and other electronic tools become available across MENA to document current patterns. It is essential, however, that consistent coding is agreed upon and used to enable intra- and inter-comparisons.

### 2.4. Antimicrobial Stewardship Programmes

ASPs have been effectively implemented in the MENA region over the past few years to enhance the appropriate prescribing of antibiotics. Checklists and guidelines have also been created among hospitals in the region and beyond to enhance the creation and implementation of ASPs despite ongoing challenges (Table 4) [55,149,150].

Encouragingly, a number of ASPs have now been undertaken among middle-income countries in MENA despite previous concerns [73], with the number likely to grow as the various countries strive to reduce rising AMR rates as part of agreed AMR NAPs.

Typically, the introduction of several interventions simultaneously to improve the prescribing of antibiotics appears to have a greater impact on future prescribing than single-activity interventions. In addition, antibiotic prescribing needs to be regularly monitored; otherwise, prescribing patterns will tend to revert back to pre-intervention levels exacerbated if there is a regular turnover of staff within the hospital.

Besides these ASPs, we have also seen clinical pharmacists giving advice regarding antibiotic prescribing in some hospitals in the MENA region, including advice on dosage adjustments and recommending more effective antibiotics. In one hospital in Oman, such activities resulted in projected net cost savings of approximately USD 200,000 per year [148]. In addition, in Jordan, among 661 blood cultures reviewed by the ASP pharmacist in a cancer center, 26% subsequently required antimicrobial therapy modification [148]. Advice included changing to a susceptible antimicrobial and the initiation of pertinent antimicrobials, as well as the discontinuation, de-escalation, and dose modification of pertinent antimicrobials—all cost-saving interventions [147]. Overall, there was a high acceptance rate of the advice at 86% among prescribing physicians.

### 2.5. Knowledge, Attitude, and Perceptions of Key Stakeholders towards Antibiotics and ASPs and Suggested Activities to Improve Future Antibiotic Prescribing in Hospitals

There are concerns with the variable knowledge of healthcare students, as well as qualified healthcare professionals, regarding their knowledge of antibiotics, AMR, and ASPs (Table 5). This matches concerns with the knowledge of antibiotics and AMR among adults in the region [163,164,165,166].

This needs to be addressed alongside other identified concerns, including prolonged prescribing of antibiotics postoperatively to prevent SSIs, to improve future prescribing, and reduce AMR among countries in the MENA region.

Suggested strategies to improve future antimicrobial prescribing among hospitals in the Middle East to reduce AMR have been divided into short to medium-term and long-term (Table 6).

## 3. Discussion

We believe this is the first comprehensive study to bring together all key aspects associated with the prescribing of antibiotics among hospitals in the MENA region to provide future guidance. This is important in the Region with such a heterogeneity in the country incomes, and concerns with growing AMR rates, reflected in ongoing NAPs and other activities to reduce AMR [10,11,12,13,14]. Such concerns will be exacerbated by high utilization of antibiotics, which has been seen among a number of hospitals in the MENA region, with Jordan having the highest rates, (up to 98% of patients studied, alongside concerns with resistance patterns [80,123,140]. This is higher than the proposed target of 40% of hospital in-patients [176].

We are aware that the published PPS studies included here were typically undertaken before the recent COVID-19 pandemic. This is an important consideration as there was typically appreciably increased prescribing of antimicrobials in patients admitted to hospitals with COVID-19 in the first wave of the pandemic across continents and countries in the absence of bacterial co-infections or secondary infections [56,57,58,177,178]. As such, this may potentially further increase antibiotic utilization and AMR rates, in addition to those documented in Table 1, without improving patient care [179,180].

Third-generation cephalosporins and penicillins, along with metronidazole, were among the most commonly prescribed antibiotics in the MENA region (Table 1). However, there are concerns with the extent of “Watch” and “Reserve” antibiotics being prescribed among hospitals in the MENA region, similar to other studies [44]. This needs urgent addressing where it occurs to reduce rising AMR rates.

There are also concerns with the extent of antimicrobials being prescribed in the pos-operative period to reduce SSIs among Middle Eastern countries, with no obvious difference between the income levels of countries in the MENA region or with other regions, including across Africa and among a number of Asian countries [63,65]. This also needs to be addressed as excessive prescribing of antibiotics will increase adverse reactions, AMR, and costs without improving patient outcomes [65]. Where necessary, setting appropriate prescription/quality targets and monitoring subsequent hospital utilization patterns as part of ASPs will improve future prescribing (Table 3). However, as mentioned, any proposed indicator must have clarity, be feasible to implement, as well as having easy-to-use, consistent, and reliable tools for valid data collection and management [60,90,91,92]. The development of appropriate indicators based on the AWaRe book and other guidance should help in this regard and address identified key issues and risk factors (Appendix A). This will be an issue in a number of hospitals in the MENA region that still rely on paper-based systems. However, this is changing and will grow as part of NAPs to reduce AMR [181,182].

A number of ASPs have been successfully introduced among MENA countries, with multiple interventions seen to have the greatest impact. There were concerns that the introduction of ASPs among LMIC MENA countries would be a concern in view of potential resource issues [73]. However, this appears to be less of an issue currently, with ASPs successfully introduced among a range of income countries in the region, providing exemplars for the future (Table 6). This is similar to the situation that now exists across Africa, also providing direction for the future [63,72,183]. It is likely that we will continue to see growth in ASPs among hospitals in the MENA region as they attempt to reduce unnecessary antimicrobial prescribing as part of NAPs to combat AMR.

As documented (Table 6), future strategies to improve the appropriateness of antimicrobial drug prescribing in the hospital sector can be divided into short-, medium-, and long-term actions. Government involvement and initiatives through NAPs are crucial to ensure sustainability, and MENA nations are currently developing and putting them into action. However, there are still several obstacles and challenges to address to undertaking agreed activities. These include available and trained staff as well as available resources and digital platforms alongside the consistent use of codes for diagnosis and other activities. We will continue to monitor the situation given rising AMR rates in the MENA region and their effects on mortality and costs.

We are aware there are a number of limitations with this paper. We did not conduct comprehensive systematic reviews for each issue, including PPS and SSI studies, as well as quality indicators and ASPs, for the reasons discussed in the Methodology section. However, as seen, we have compiled a comprehensive list of PPS and SSI studies currently undertaken among hospitals in the MENA region, as well as a range of prescribing and quality indicators that have been used in practice. Alongside this, we have documented a number of ASPs that have been successfully implemented across a range of countries in the Middle East as exemplars. Consequently, despite these limitations, we believe our findings, suggestions, and conclusion are robust, given the number of examples combined with our methodology.

## 4. Materials and Methods

The principal strategy involved a narrative review of key areas to comprehensively inform current antimicrobial utilization patterns across hospitals in the MENA region, including concerns and potential ways forward to address key issues. These include the instigation of agreed prescribing and quality indicators as part of planned ASPs. The co-authors’ extensive expertise in working with patients with infectious diseases, documenting current utilization patterns, adopting policies to improve future prescribing, which includes the creation of applicable quality indicators, and researching and implementing ASPs, have added to this. We have used this approach before when debating key areas across multiple countries and continents, including key issues and challenges surrounding infectious diseases [45,63,65,184,185,186,187].

### 4.1. Current Antimicrobial Utilisation Patterns among Hospitals across the MENA Region

The methodology built upon a recent systematic review of PPS studies undertaken by some of the co-authors [40], and subsequently involving studies from 2016 onwards until October 2022. 2016 was chosen as this was the launch of the WHO Global Action Plan to reduce AMR [93]. This methodology was employed since we were aware that a number of possible PPS studies would not be listed in databases, including PubMed and Web of Science, building on recent experiences [63]. However, we wanted to include them as our intention was to comprehensively document a range of PPS studies, and their findings, across the MENA countries to provide a baseline for future studies. We also purposely did not select which countries from the MENA region to include in this narrative overview in order not to bias any findings.

Similar to the systematic review of Saleem et al. (2020) and the recent Pan-African study [40,63], key categories included the number of participating hospitals within the PPS study, the PPS methodology, e.g., ECDC, WHO, or Global PPS, [27,112,124,176], as well as the first, second, or third most prescribed antibiotic broken down by ATC code and the AWaRe classification [47,48,49,188]—the latter, especially with growing concerns regarding the extent of prescribing of “Watch” and “Reserve” antibiotics among a number of Western and Central Asian countries [44,47,50]. In addition, whether antibiotics were prescribed for prophylaxis or treatment and the average number of antibiotics prescribed per patient.

As mentioned, we excluded the findings from the two recent Global PPS studies, e.g., Versporten et al. (2018) and Pauwels et al. (2021), since it was difficult to pull out individual hospital data to populate the respective Tables [44,86]. However, the findings from these two studies were discussed in relation to the findings from the various MENA countries.

The various MENA countries were broken down by their World Bank classification, i.e., low-income, low-middle, upper-middle income, and high-income countries, building on the recent study of Adekoya et al. (2021), as well as the recent Pan-African study for consistency [63,131].

### 4.2. Antibiotic Prophylaxis to Prevent Surgical Site Infections

The principal approach was a narrative review, which built upon recent publications involving some of the co-authors [65,189]. This was supplemented by additional studies from 2016 onwards known to the co-authors. This is similar to the approach adopted by the authors in other studies, including the recent Pan-African study [40,63]. In addition, a narrative review of ongoing concerns with the management of SSIs among Middle Eastern countries provides a background to the importance of this area for future quality improvement initiatives.

The various MENA countries were again broken down by their World Bank classification, i.e., low-, low-middle, upper-middle, and high-income countries, building on the recent study of Adekoya et al. (2021) and the Pan-African study for consistency [63,131].

### 4.3. Prescribing and Quality Indicators

The principal approach was a narrative review. This built upon recent PPS and ASP publications from across the MENA area, supplemented with additional studies known to the co-authors. This mirrors the approach for the PPS and SSI studies.

### 4.4. Antimicrobial Stewardship Programs and Subsequent

Again, the principal approach was a narrative review of recent ASPs that had been instigated among hospitals across the Middle East, as well as published papers on the knowledge, attitude, and practices among key stakeholders towards antibiotics and AMR. This includes details of the interventions undertaken from 2016 onwards as well as the outcomes against agreed indicators.

The MENA countries that had instigated ASPs were again broken down by their World Bank classification, i.e., low-, low-middle, upper-middle, and high-income countries, building on the recent study of Adekoya et al. (2021) and the Pan-African study for consistency [63,131]. This is important as there have been concerns about conducting ASPs in LMICs due to issues of available and trained personnel as well as the necessary finances [73].

Possible short to medium and long-term activities to improve future antibiotic prescribing in hospitals have been based on the findings from the various narrative reviews combined with the considerable knowledge of the co-authors, building upon similar activities across Africa [63].

## 5. Conclusions

In conclusion, reducing AMR must be a high priority for all MENA countries, with AMR potentially developing into the next pandemic unless it is addressed. However, in order to lower AMR rates, numerous coordinated efforts need to be undertaken as part of agreed AMR NAPs. This calls for better awareness of current antimicrobial utilization patterns in hospitals, along with key targets for quality improvement programs.

The study identified concerns with current antibiotic prescribing in hospitals, including prolonged administration to prevent SSIs as well as the extent of inappropriate prescribing, including “Watch” and “Reserve” antibiotics. Greater use of the AWaRe classification and guidance will help in this regard, along with other suggestions for the future. We will continue to monitor these developments to enhance future appropriate prescribing of antimicrobials in hospitals throughout the MENA region. This is crucial given the lack of novel antimicrobials being produced.

## Figures and Tables

**Figure 1 antibiotics-12-00827-f001:**
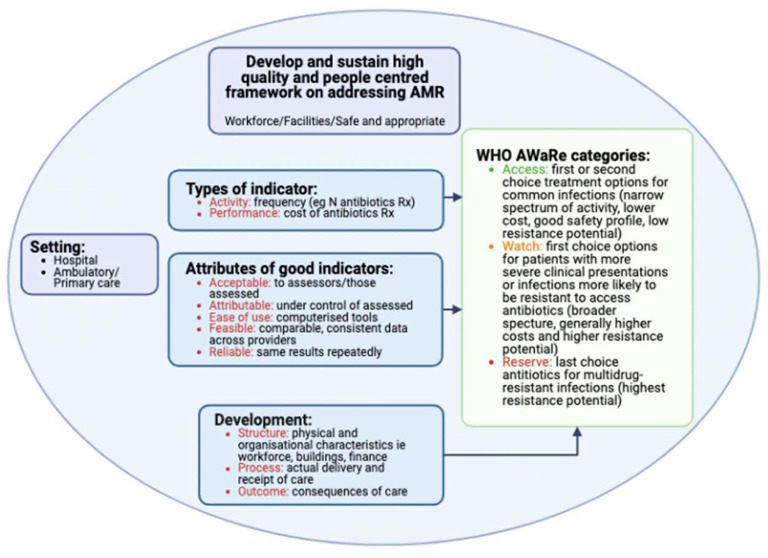
Key principles for indicator development for antimicrobials across the MENA region (first published for African countries; reproduced with the kind permission of the authors [63]).

**Table 1 antibiotics-12-00827-t001:** Summary of PPS studies.

Country	Author and Year	No. of Hospitals	PPS Timing (Period)	PPS Protocol	Study Duration (Period)	AM Use Rate n (%)	1st AM ATC Code (%) and AWaRe Classification	2nd AM ATC Code (%) and AWaRe Classification	3rd AM ATC Code (%) and AWaRe Classification	Used for Prophylaxis(%)	Used for Treatment (%)	Total Antimicrobials Delivered (Per/Patient) for Their Infections
**Low Middle-Income Countries ***												
Egypt	Ashour et al., 2022 [112]	1	Daily two weeks	ECDC	During July 2019	300 (79.15%)	Aminopenicillins+ Beta-lactamase inhibitors (A) *J01CR* (43.3%)	3rd Generation cephalosporins (W) *J01DD* (29.0%)	Fluroquinolones (**W**) *J01MA* (8.7%)	62.3	37.7	1.9
Egypt/Various	Alothman et al., 2020 [52]	11	One day	ECDC	3 April 2018	1586 (84.9%)	Cephalosporins (W) *J01D* (32.4%)	Carbapenems (W) *J01DH* (18.8%)	Glycopeptides (W) *J01XA* (16.9%)	-	28.3	-
Iran	Fahimzad et al., 2016 [113]	16	Daily over one week	Standard	January 2014 and February 2014	571 (66.66%)	Ceftriaxone (**W**) *J01DD04* (36.6%)	Metronidazole (**A**) *J01XD01* (23.8%)	Vancomycin (**W**) *J01XA01* (18.8%)	26.77	73.1	-
Iran	Soltani et al., 2018 [114]	2	Two weeks	ARPEC-web PPS	October 2011 and 2012	252 (64%)	Ceftriaxone (**W**) *J01DD04* (19.9%)	Ampicillin (**A**) *J01CA01* (14.3%)	Vancomycin (**W**) *J01XA01* (13.3%)	7.5	88	-
Morocco	Razine et al., 2012 [115]	1	-	-	January 2010	392 (32.8%)	Amoxicillin Clavulanic acid (**A**) *J01CR02* (32%)	3rd Generation Cephalosporins (W) *J01DD* (13%)	Gentamycin (**A**) *J01GB03* (10%)	-	-	-
Morocco	Chiguer et al., 2018 [116]	1	-	ECDC	5 June to 19 July 2017	-	3rd generation cephalosporins (W) *J01DD* (38.2%)	-	-	-	-	-
Palestine	Alagha et al., 2022 [117]	3	-	WHO	July 2019 to January 2020	1400 (68.2%)	Ceftriaxone (**W**) *J01DD04* (47.5%)	Cefazolin (**A**) *J01DB04* (10.6%)	Ciprofloxacin (W) *J01MA02* (**W**) (8.6%)	-	-	1.26
Tunisia	Ayed et al., 2019 [118]	2	One Week	Self	3–10 July 2017	371 (49.2%)	-	-	-	20	80	-
Tunisia	Maamri et al., 2021 [119]	2	-	-	2019	410 (39.2%)	Penicillin (A) *J01C* (26.7%)	Fluroquinolones (W) *J01MA* (20%)	3rd generation cephalosporins (W) *J01DD* (16.5%)	24.9	-	-
**Upper Middle-Income Countries ***												
Iraq	Nassr et al., 2018 [53]	4	Period	Descriptive statical analysis	October 2017 and April 2018	177 (88.5%)	Parenteral 3rd generation cephalosporins (W) *J01DD* (54.3%)	Nitroimidazole *P01AB* (29.4%)	Quinolones (W) *J01M* (7.5%)	-	-	1.65
Iraq	Kurdi et al., 2020 [27]	3	Period	Global PPS	Sep-Dec 2019	192 (93.7%)	Cephalosporins 3rd generation *J01DD* (52.6%)	Imidazole derivatives *J01XD* (16.9%)	Beta-lactam Penicillin *J01C* (10.5%)	17.3	82.7	1.38
Iraq	Kurmanji et al., 2021 [120]	1	Period	Global PPS	January–April 2019	402 (66.2%)	Ceftriaxone (**W**) *J01DD04* (70%)	-	-	-	-	-
Iraq	Kurmanji et al, 2021 [121]	5	Period	Global PPS	January 2019 to April 2019	808 (66.7%)	Ceftriaxone (**W**) *J01DD04* (31.5%)	Metronidazole (**A**) *J01XD01* (20.9)	Meropenem (**W**) *J01DH02* (10.1)	51.1	42.4	-
Jordan	Elhajji et al., 2018 [122]	1	One day	Global PPS	2015	85(78.2%)	3rd generation Cephalosporins (W) *J01DD* (26.2%)	Fluroquinolones (**W**) *J01MA* (18.5%)	Carbapenems (R) *J01DH* (15.4%)	40	60	1.42
Jordan	Abu Hannour et al., 2020 [123]	1	Period	Global PPS	June–July 2018	488 (98.1%)	Cephalosporins *J01D* (50.6%)	Carbapenems (R) *J01DH* (39.6%)	Imidazole derivatives (A) *J01XD* (22.4%)	-	-	1.57
Jordan	Ababneh et al., 2021 [124]	1	One day	Standard	13 August 2018	144 (21.1%)	-	-	-	-	-	-
**High-Income Countries ***												
Bahrain	Al Salman et al., 2017 [125]	1	-	Global PPS	1 Feburary 2015 to 30 April 2015	263 (70.7%)	Beta-lactam Other than penicillin (W) *J01D* (42.5%)	Penicillin (A) *J01C* (15.3%)	-	-	-	-
Qatar	Saleem et al., 2020 [40]	1	Repeated	ECDC	April–May 2012	25 (43.0%)	Penicillin plus beta-lactamase inhibitors (W) *J01CR* (39.4%)	Carbapenems (RW) *J01DH* (15.2%)	Fluroquinolones (**W**) *J01 MA* (9.1%)	6.1	93.9	1.32
Saudi Arabia	Al Matar et al., 2018 [126]	26	One Day	Global PPS	May 2016	2182(46.9%)	3rd generation cephalosporins *J01DD* (17.2%)	Penicillin *J01C* (9.4%)	Penicillin and enzyme inhibitor *J01CR* (8.8%)	34.6	61.4	1.4
Saudi Arabia	Alahmadi et al., 2020 [127]	2	Two-week period	ECDC	September 2016 and November 2016	332(49.18%)	3rd generation cephalosporins (W) *J01DD* (16.5%)	2nd generation cephalosporins (W) *J01DC* (8.6%)	Fluroquinolones (**W**) *J01MA* (8.6%)	-	-	1.5
Saudi Arabia	Al-Tawfiq et al. [128]	1	Period	-	January 2017 to January 2019	200 (40%)	Meropenem (**W**) *J01DH02* (18%)	Cefazolin (**A**) *J01DB04* (10%)	Ceftriaxone (**W**) *J01DD04* (8%)	8.5	89.5	1.3
Saudi Arabia	Haseeb et al., 2021 [54]	6	Period	Global PPS	January 2019 to July 2019	447 (61.9%)	Ceftriaxone (**W**) *J01DD04* (15%)	Piperacillin (**W**) *J01CA12* (10.9%)	Metronidazole (**A**) *J01XD01* (7.1%)	7.9	92.1	1.7
Saudi Arabia	Alsaedi et al., 2022 [129]	6	One day	Standard	2017	240(14.4%)	Carbapenems (R) *J01DH* (19.6%)	Cephalosporins *J01D* (14.8%)	Vancomycin (**W**) *J01XA01* (13.2%)	-	-	-
United Arab Emirates	Alnajjar et al., 2022 [130]	1	One day	ESAC	26 January 2020	41(32.8%)	Combinations of penicillin (W) *J01CR50* (31.5%)	Amoxicillin-clavulanic acid (**A**) *J01CR02* (22.2%)	Piperacillin-tazobactam (**W**) *J01CR05* (9.3%)	29.6	70.4	1.3

NB: AM: antimicrobial; ATC: anatomical, therapeutic, chemical classification; AMU: antimicrobial use; A and W: Access and Watch (AWaRe classification) [46,47]; Period: period of the study if either over several months or within a month with no further details given; Proph: prophylaxis; Treat: treatment; * World Bank Classification based on [131].

**Table 2 antibiotics-12-00827-t002:** Length of prescribing of antibiotics postoperatively among hospitals in the MENA countries to prevent SSIs.

Country	Author and Year	Findings including Number of Patients Where Documented
**Low Middle-Income Countries ***		
Egypt	Ashour et al., 2022 [112]	98.5% of patients (134/136) were prescribed antibiotics for >24 h for SAP, with no patient prescribed single-dose prophylaxis preoperatively; Median duration of antibiotic use was 3 days per patient (range, 2–9 days).
Tunisia	Ayed et al., 2019 [118]	Out of 371 admitted patients, 73 (20%) were administered, with a duration > 1 day.
**Upper Middle-income Countries ***		
Iraq	Nassr et al., 2018 [53]	All patients (77) undergoing SAP were prescribed antibiotics >1 day.
Iraq	Kurmanji et al., 2021 [120]	89% of patients received SAP for more >1 day for the different surgical categories, including obstetric and gynecological indications.
Iraq	Kurnanji et al., 2021 [121]	65.5% of surgical and medical prophylactic antibiotics were used for >1 day, especially ceftriaxone, across five teaching hospitals.
Jordan	Abu Hammour et al., 2020 [123]	41 patients (adult and children) were prescribed overall 46 antibiotics for SAP; 8.7% were prescribed a single dose, 30.4% more than one dose for one day, and 60.9% for more >1 day.
**High-Income Countries ***		
Saudi Arabia	Al Matar et al., 2019 [126]	Antibiotics for SAP represented 23.4% of total antibiotic doses (758/3214); 78% of doses for SAP were administrated for >24 h; only 10% of doses were given as a single dose.
Saudi Arabia	Al-Tawfiq et al., 2020 [128]	Duration of antibiotic administration for SAP was typically >1 day.
Saudi Arabia	Haseeb et al., 2021 [54]	20.9% of patients undergoing SAP received a single dose of antibiotics, 35.2% for one day and 43.9% for >1 day.
United Arab Emirates	Alnajjar et al., 2020 [130]	10/16 patients (62.5%) received antibiotics for more than one day to prevent SSIs.
United Arab Emirates	Alshehhi et al. 2021 [67]	Antibiotics for SAP were typically administered for 3 days (104 patients; 32.9%), 5 days (89 patients; 25.7%), and 7 days (41 patients; 11.8%) prior to an ASP; This reduced to 3 days (118 patients; 80.8%); 5 days (10; 6.8%) and 7 days (6; 4.1%) following the ASP; however, still an appreciable number were non-compliant with current guidelines; The average duration of cefuroxime (most prescribed antibiotic) was 3.8 ± 1.2 days pre-ASP—reducing to 2.6 ± 0.6 days post-ASP.
United Arab Emirates	Vippadapu et al., 2022 [68]	Almost all patients undergoing SAP to prevent SSIs were prescribed discharge antimicrobials (99%)

NB: ASP: Antimicrobial Stewardship Program; SAP: surgical antibiotic prophylaxis; SSIs: surgical site infections; * World Bank Status (Based on [131]).

**Table 3 antibiotics-12-00827-t003:** Indicators that have been used among in-patients in hospitals across Middle East Countries to assess the prescribing of antibiotics.

Indicator	Reference
** *Activity (Process/Performance) Indicators* **	
Defined daily doses (DDDs) and DDDs/100 or 1000 bed-days/patient days	[124,133,134,135,136,137,138,139,140]
% of inpatients prescribed antibiotics/antimicrobials	[27,40,86,112,114,125,126]
Days of therapy per 1000 study patient days	[70]
Average number of antibiotics prescribed per in-patient	[53]
% of patients’ notes where the indication for administering antibiotics is documented/stop or review dates recorded	[27,53,86]
% of patients prescribed antibiotics postoperatively for SAP for longer than 24 h after surgery/duration of SAP postoperatively	[67,123,126,134]
% Empiric prescribing vs. targeted antibiotic prescribing (following culture and sensitivity testing results)	[27,86,113,114,141]
% of “Access” versus “Watch” or “Reserve” antibiotics or % reduction in targeted antibiotics	[44,82,137,142,143]
% decreased prescribing of restricted antibiotics	[144,145,146]
% compliance with current guidelines	[54,86,126,134,139,142]
% de-escalation, including IV to oral switches and their timing	[82,133,139,142,147]
% reduction in length of stay	[67,141]
% reduction in expenditure of (targeted) antimicrobials	[137,138,143,146,148]
** *Outcome indicators* **	
% of patients postoperatively getting SSIs	[60,61,62]
% reduction in resistance rates to targeted pathogens	[135,140,145,146]
% reduction in re-admission rates	[82]
% decrease in healthcare-associated infections	[138,139,146]
% reduction in mortality rates	[82,136,141,142,144]

NB: AWaRe classification for antibiotics—Access, Watch, or Reserve [46,47]; IV: intravenous; SAP: surgical antibiotic prophylaxis; SSIs: surgical site infections.

**Table 4 antibiotics-12-00827-t004:** Summary of published studies among MENA countries documenting ASPs and their impact.

Country	Author, and Year	Intervention and Aim	Impact of the Intervention
**Lower Middle-Income Countries ***			
Iran	GolAli et al., 2017 [142]	Over a nine-month period, the medical files of all patients who received IV antibiotics were reviewed by a clinical pharmacy specialist; Discrepancies from international and local guidelines were discussed with physicians to improve their prescribing.	198 antibiotic courses were reviewed; 167 recommendations were made with an acceptance rate of 80.2%. Improvements made post-intervention resulted in: ○ 47% improvement in the choice of antibiotic (*p* < 0.001) and de-escalation (48%, *p* < 0.001); ○ On-time changing from IV to oral therapy (60%, *p* < 0.001) and dosing schedules (30%, *p* = 0.003); ○ Significant reduction in hospital length of stay from 16.1 days to 11.6 days (*p* < 0.05); ○ Reduced trend in the prescribing of carbapenems (**W**), vancomycin (**W**), and ciprofloxacin (**W**).
Iran	Rahbarimanesh et al., 2019 [141]	Over a 6-day period each week, the ASP team, including ID physicians and pharmacists, inspected the primary instructions and applied daily and weekly dosing patterns for meropenem and vancomycin; These antibiotics were purposefully selected for concerns with high prescribing rates; In addition, the ID pharmacist notified key physicians treating patients of clinical microbiology laboratory findings.	135 children were included in the ASP; The levels of antimicrobial prescribing, dosing, and duration of antimicrobials prescribed were significantly lower after the ASP; The length of hospital stay was also significantly lower after the ASP (15.6 ± 2.8 vs. 22.7 ± 1.9).
Iran	Mahmoudi et al., 2020 [137]	In May 2017, an ASP team consisting of 2 infectious disease (ID) specialists, 2 clinical pharmacists, a hospital administrator, a microbiologist, and an information technology specialist was formed; The combined intervention included guideline revisions and the development of “antimicrobial order forms”, information and education of key groups, regular ward rounds, and intensified ID consultations and feedback.	94,000 prescriptions were analyzed over a 2-year period; Antimicrobial consumption measured in DDD/1000 bed days dropped by 24.8%, 25.0%, 35.3%, 47.0%, 39.2%, 10.5%, and 23.2% for amphotericin B, caspofungin, colistin, voriconazole, meropenem, imipenem, and vancomycin, respectively; Linezolid consumption increased by 26.8% after implementing the ASP; Expenditure of target antimicrobials decreased by 41.3% after the intervention (= 0.001). Increased antimicrobial susceptibility of *Pseudomonas aeruginosa* after the intervention (amikacin, carbapenems, cefepime, ciprofloxacin, colistin, and piperacillin/tazobactam—typically **W** and **R** antibiotics).
Iran	Hajiabdolbaghi et al., 2020 [151]	Reviewing patients from two internal disease wards in a tertiary center via ID consultations; ID specialists were also asked to confirm carbapenem use in patients within 48 h of initiation; Understanding that ASPs help prevent AMR through improved utilization of antimicrobial agents.	Of the 186 consultations conducted by the ID specialists, 28 (15%) consultations led to an antibiotic change, 46 (25%) led to antibiotic discontinuation, and 112 (60%) carbapenems were continued; Crude mortality rates in the two internal medicine wards were calculated to be 2.6%—with no significant change compared to the previous year (CMR: 2.9%) despite the changes, including discontinuation.
Iran	Bahrampour Juybari et al., 2022 [152]	ASP introduced to enhance compliance to imipenem and meropenem administration guidelines within the hospital; Compliance assessed with a developed checklist including six items.	6,032 imipenem/meropenem vials (1 g/vial) were prescribed during the study (meropenem for 210 patients and imipenem for 87 patients); In 64.2% of the patients, there was no indication for the antibiotic prescribed, and the mean score of the subjects was 1.55 ± 1.2 out of 6; The obtained score was 3 out of 6 in 53 (17.8%) records and 4 out of 6 in 18 (6.1%) records; The mean ASP score in the ICUs was higher, while it was lower in the surgical ward as compared to the other wards (*p* = 0.002); Greater intensity of education and monitoring post-implementation are necessary as part of ASPs to improve future prescribing habits.
Jordan/Various including Yemen	Bhalla et al., 2016 [143]	An ASP was introduced in this humanitarian surgical hospital to improve antimicrobial use—especially surrounding resistant organisms; Antibiotics were reviewed with real-time recommendations made by the team to optimize choice, dose, and duration.	An overall reduction in the use of broad-spectrum antibiotics; The cost of antibiotics for the surgical hospital declined considerably from approximately USD 252,077 (average, USD 21,006/month) pre-ASP to < USD 159,948 (USD 13,329/month) post-ASP.
Jordan	Yusef et al., 2021 [145]	Introduction of an ASP consisting of several components: ○ education, introduction, and dissemination of antibiotic guidelines; ○ antibiotic restriction policies with prior approval for target antibiotics; ○ audit of compliance with the restriction policy and feedback.	There was strong evidence of decreased antibiotic use post-ASP in targeted antibiotics, including: imipenem/cilastatin (*p* = 0.0008), all carbapenems (*p* = 0.0001), third-generation cephalosporins (*p* = 0.0004), vancomycin (*p* = 0.0006), and colistin (*p* = 0.0016); A strong association in terms of a decrease in the slope of antibiotic use post-ASP for ertapenem (*p* = 0.0044) and ciprofloxacin (*p* = 0.0117); There was an increasing trend in the prescribing of piperacillin/tazobactam pre-ASP, with this trend halted post-ASP (*p* = 0.4574); The ASP was associated with reduced levels of carbapenem-resistant *Acinetobacter baumannii* (*p* = 0.0237).
Jordan	Khasawneh et. al., 2021 [153]	To assess Jordanian practitioners’ perceptions and practices towards ASPs; A close-ended structured questionnaire comprising 34 questions.	286 participants were enrolled in the study; There was an overall positive perception toward ASPs (measured using a structured questionnaire comprising 34 items); Long years of practice, postgraduate studies, and practice in academic sectors yielded higher perception scores (*p* = 0.0335, 0.0328, and 0.0007, respectively).
Jordan	Darwish et al., 2022 [140]	Introduction of AMS teams in two hospitals consisting of ID physicians, clinical pharmacists, microbiology personnel, infection personnel; The AMS team also included infection control nurses, as well as quality control department staff in one hospital and the IT director in another; Restriction of antibiotic use and other strategies were introduced as part of the ASPs, as well as the introduction of local guidelines.	Antibiotic susceptibility testing to appropriate antibiotics improved in both hospitals; The incidence of extended-spectrum beta-lactamase (ESBL) positive *E. coli*, ESBL positive *Klebsiella*, and vancomycin-resistant Enterococcal species decreased numerically post-ASP while methicillin-resistant *Staphylococcus aureus* showed an increase in incidence during the second year of the study, although there was no change in ASP adherence; Overall, the study emphasized the positive impact of ASPs; however, more can be undertaken to improve prescribing.
**Upper Middle-Income Countries ***			
Lebanon	Chamieh et al., 2019 [135]	The ASP, ID team, and ICU physicians approved a plan to reduce empiric prescribing of carbapenems as well as colistin, tigecycline, or both for patients confirmed with, or at high risk, of *A. baumanii* infections; ID physicians evaluated the clinical severity and hemodynamic stability of each patient and had the final discretion to prescribe either colistin or tigecycline.	536 patients were involved during the study period (ICU) over a one-year period starting 1 February 2016; Carbapenem consumption decreased by 59% during the study period; Overall restricted antimicrobial drug consumption, including carbapenems, carbapenem-sparing regimens, colistin, and tigecycline (**W** and **R**) dropped 637 DDD/1000 patient-days (*p* < 0.005); Isolation of *A. baumanii* decreased substantially, additionally, colistin utilization decreased 55% from 20 DDD/1000 patient-days in period 1 to 9 DDD/1000 patient-days in period 2 (*p* = 0.019)—pre and post-intervention; Tigecycline consumption remained unchanged; 78% decrease of *A. baumanii* isolated in sputum and near-elimination of *A. baumannii* carrying the *bla*_oxa-23_ gene (all of which were multilocus sequence type 2 clone) over the 1-year study period.
Lebanon	Moghnieh et al., 2020 [136]	Implementation of an ASP based on the “handshake” strategy for 2 years in this hospital.	The mean use density levels for imipenem and meropenem decreased by 13.72% (*p* = 0.017), coupled with a decreasing rate of prescribing (−24.83 DDD]/1000 patient days /month; *p* = 0.02); Reduction in the use of colistin (−5.51 DDD/1000 PD/month = −5.5; *p* = 0.67); Decrease in costs by 24.6% after ASP implementation (*p* < 0.0001); Decrease in nosocomial bacteremia caused by carbapenem-resistant Gram-negative bacteria by 34.84% (*p* = 0.13); Noticeable reduction in the incidence rate of bacteremia due to carbapenem-resistant *Acinetobacter baumannii* (−0.24 cases/1000 PD/month, *p* = 0.01).
Lebanon	Sayegh et al., 2021 [154]	Descriptive cross-sectional survey undertaken using an online questionnaire; The survey items assessed ASP implementation, as well as physicians’ attitudes, usefulness of ASP tools, and barriers to ASP implementation.	158 physicians completed the survey—although a low response rate (4%); The majority (66%) of physicians were familiar with the ASP concept; Overall positive attitudes and support for ASP implementation with antibiotic ward rounds as well as prospective audit coupled with feedback rated as most useful ASPs; However, ASPs were reported as affecting physicians’ autonomy by 34% of participants; Barriers included: ○ A lack of regular educational programs (41%); ○ Lack of support from the medical staff (76%); ○ Minimal support from the Ministry Of Public Health for ASPs; ○ Absences of regulations and national guidelines; ○ Shortage of infectious disease physicians.
Lebanon	Shallal et al., 2022 [70]	During the intervention phase, the AMS team reviewed antimicrobial use within 72 h post-prescription, as well as providing recommendations based on current guidelines; The dedicated AMS team consisted of an ID specialist, as well as infection prevention and control officers.	The total days of therapy decreased from 11.46 days during the baseline phase to 8.64 days during the intervention phase (*p* < 0.001); After adjusting for age, sex, and treatment indication, patients in the pre-intervention phase were prescribed antibiotics 29% longer (*p* < 0.001) and hospitalized 16% longer (*p* < 0.001) than patients in the intervention phase; Intervention acceptance by the physicians occurred 88.5% of the time.
**High Income Countries ***			
Qatar	Garcell et al., 2017 [134]	The intervention consisted of the education of the surgical staff regarding appropriate prophylaxis to prevent SSIs, including optimal duration of administration; Monitoring and feedback of subsequent antibiotic prescribing to prevent SSIs.	603 patients were included; Timely administration of antibiotics for SAP was achieved in 72.9% of the procedures during 2013—with an increase to 99.6% and 100% (*p* < 0.001) in 2014 and 2015, respectively; Antibiotics were prescribed as a single prophylaxis in 61.5% of cases overall—the highest level in 2015 (*p* < 0.05); Compliance with discontinuation criteria increased from 86.4% (2013) to 92.2% (2014) and 96.7% in 2015 (*p* < 0.05); Most frequent antimicrobials prescribed were cefuroxime (**W**), metronidazole (**A**), and ceftriaxone (**W**)—accounting for >90% of doses—with a sustained reduction in cefuroxime—26.2% lower in 2015 compared with 2013 (*p* < 0.05).
Qatar	Nasr et al., 2019 [155]	Feedback intervention from clinical pharmacists combined with qualitative feedback; Documenting changes in improved patient documentation, including diagnosis.	Educational sessions and awareness campaigns involving multidisciplinary teams on a regular basis to improve antibiotic use and prescribing in hospital settings, requiring rigorous efforts from all team members for maximum impact; Focus-group educational sessions have a greater impact than on-the-spot educational interventions during multidisciplinary rounds as teams change on a regular basis, and it is better to educate all groups at once; Complete patient documentation increased following clinical pharmacists’ interventions.
Qatar	Shaukat et al., 2020 [139]	Multidisciplinary AMS team comprising ID physician, clinical pharmacists, clinical microbiologists, and infection control practitioners; Interventions included (i) formulary restrictions, (ii) hard stop of restricted antimicrobials after 48 h by pharmacy personnel, (iii) analysis of clinical interventions and feeding back to prescribers, and (iv) IV to oral conversion where appropriate.	2500 ASP interventions undertaken between January 2018 and December 2019; Overall, the appropriateness of antimicrobial prescribing (according to local guidelines) improved over time—enhanced by regular monitoring/auditing, continuous education, and increasing awareness among prescribers Discontinuation and de-escalation rate of restricted antimicrobials was cumulatively on approximately 60% of occasions, with only 35–40% of restricted antimicrobials continued; 40% improvement in IV to oral switches post-ASP; 50% reduction in hospital-associated *Clostridium difficile* cases.
Qatar	Sid Ahmed et al., 2020 [156]	A formal ASP was introduced by a team consisting of an ID physician and a pharmacist; The principal strategy of the ASP team was a prospective-audit-and-feedback approach; There was a pre-authorization requirement for all broad-spectrum antibiotics with a principal target of reducing the prevalence of multi drug resistant (MDR) *Pseudomonas aeruginosa*.	6501 clinical isolates of *P. aeruginosa* were collected prospectively over a 3-year period starting October 2014; The prevalence of MDR *P. aeruginosa* decreased from 9% (166/1844) in 2015 (pre-ASP) to 5.46% (122/2234) post-ASP (*p* = 0.019); Alongside this, a 23.9% reduction in studied antimicrobials (b-lactam/b-lactamase inhibitor combinations, third and fourth-generation cephalosporins, glycopeptides, tigecycline, all carbapenems, and all fluroquinolones—**W** and **R**), post ASP implementation (*p* = 0.008); This included a 32.6% reduction in the prescribing of meropenem.
Saudi Arabia	Amer et al., 2013 [157]	Compare prescribing appropriateness rates before and after the implementation of an ASP in a tertiary care hospital.	73 subjects were recruited, 49 in a historical control and 24 in the active arm; The appropriateness of empirical antibiotics improved from 30.6% (15/49) in the historical control arm to 100% (24/24) in the proactive ASP arm (*p*-value < 0.05); For the ASP group, initially, 79.1% (19/24) of antibiotic prescribing was inappropriate; This diminished following the implementation of the ASP to 0% on the implementation of the recommendation; 27 interventions were made with an acceptance rate of 96.3%.
Saudi Arabia	Alawi et al., 2016 [144]	An AMS team was formed to: Supervise antibiotic use by verifying the appropriateness of all prescriptions, using the findings from culture and sensitivity testing and clinical data; Restrict the use of certain antibiotics; Ensure compliance with peri-operative SAP.	67.2% decrease in the dispensing of restricted antibiotics from 17,763 dispensed units/month pre-ASP to 5819 post-ASP; Decreasing incidence trend across all monitored infections—the only strong association was seen for *Acinetobacter baumannii* (*p* = 0.007); Decreased expenditure on restricted antibiotics—average monthly saving up to USD 326,020; The mean mortality rate decreased from 90.83 (±10.12) deaths/10,000 patient months pre-ASP to 75.25 (±17.46) post-ASP.
Saudi Arabia	Momattin et al., 2018 [133]	ASP instigating targeting IV to oral conversion program, a vancomycin pharmacokinetic program, antibiotic de-escalation, and preoperative antibiotic protocols for SAP utilizing adapted orders, and a multi-faceted approach to decrease prescribing of antibiotics for respiratory tract infections; Antibiotic usage data were collected for 2011 as a baseline and compared to 2013, 2014, and 2015.	Total reduction in antibiotic usage after adjustment using 100 bed-days and a case-mix index (CMI) was 5.13% between the baseline and the latest year, i.e., from 95.6 to 90.7 between 2011 and year 2015; Adjusting by CMI is important when benchmarking antibiotic utilization dates between hospitals; An earlier published study showed that ASP resulted in a higher percentage of patients achieving a therapeutic range when vancomycin was prescribed alongside less nephrotoxicity [158].
Saudi Arabia	Baraka et al., 2019 [159]	To investigate practitioners’ perceptions regarding AMS implementation and to identify challenges and facilitators of these programs’ execution; More than 50% of clinicians (N = 184) reported a lack of awareness of AMS programs and their components, whereas 71.2% did not have previous AMS experience.	Lack of administrative awareness about AMS programs; lack of personnel, time limitation, limited training opportunities, lack of confidence, financial issue or limited funding and lack of specialized AMS information resources were reported 65.8%, 62.5%, 60.9%, 73.9%, 50%, 54.3%, and 74.5%, respectively to limit their introduction; 77.2% of respondents reported that formulary management could be helpful for AMS practices, and the majority of the respondents (79.9%) reported that the availability of pathogens and antimicrobial susceptibility testing could be helpful for ASPs.
Saudi Arabia	Haseeb et al., 2020 [160]	Aims to identify ASPs in Makkah region hospitals and their perceived level of success; Explore current progress and issues related to the implementation of ASPs in Makkah region hospitals at the pharmacy level through a survey (n = 25).	Among responding hospitals, 19 (76%) reported that the most commonly reported ASPs were: ○ formulary restrictions (90%) for broad-spectrum antimicrobials; ○ use of prospective feedback on antimicrobial prescribing (68%); ○ use of clinical guidelines and pathways (100%); ○ use of automatic stop orders (68%) to limit inappropriate antimicrobial therapy. All reported institutions had at least one ASP with a high success rate; Overall, a multidisciplinary approach, active drug and therapeutic committees, and educating pharmacists and physicians on ASPs were important for successful ASPs in this region.
Saudi Arabia	Haseeb et al., 2021 [161]	The AMS algorithm (Start Smart and Then Focus) and an ASP toolkit were distributed to all intensive care unit staff; The aim was to assess the implementation of a structured multidisciplinary ASP in a critical care situation by promoting the judicious prescribing of antimicrobials.	Multidisciplinary ASP in a 20-bed critical care setting was conducted; The defined daily dose (DDD) per 100 bed days was reduced by 25% (742.86 vs. 555.33; *p* = 0.110) compared to 7% in the control condition (tracer medications) (35.35 vs. 38.10; *p* = 0.735); Use of IV ceftriaxone (**W**) measured as DDDs per 100 bed days was decreased by 82% (94.32 vs. 16.68; *p* = 0.008), whereas oral levofloxacin use was increased by 84% (26.75 vs. 172.29; *p* = 0.008) in the intensive care unit; The multidisciplinary ASP group, daily audits, and feedback by clinical pharmacists and physicians with infectious disease training alongside continuous educational activities about antimicrobial use and AMR and the use of local antimicrobial prescribing guidelines improved antibiotic use in this institution.
Saudi Arabia	Al-Omari et al., 2020 [138]	The ASP included: Education of prescribers and other healthcare workers; Audit and feedback activities—real-time, either written, oral, or retrospective; Restrictive interventions, including pre-authorization of targeted antibiotics.	409,403 subjects were recruited—79,369 pre-ASP and 330,034 post-ASP; DDDs fell for all targeted broad-spectrum antimicrobials (imipenem/cilastatin, piperacillin/tazobactam, colistin, tigecycline, cefepime, meropenem, ciprofloxacin, moxifloxacin, teicoplanin, and linezolid—**W** and **R**); from 320 DDDs/1000 patient-days to 233 post-ASP Antimicrobial expenditures decreased by 28.45% in the first year of the program and remained relatively stable in subsequent years; Rates of hospital-acquired infections (HAIs) involving *C. difficile*, ventilator-associated pneumonia (VAP), and central line-associated bloodstream infection (CLABSI) decreased after the ASP; Incidence of HAIs in 2015 vs. post ASP—*C. difficile*, 94 vs. 13, *p* = 0.024; VAP, 24 vs. 6, *p* = 0.001 and CLABSI, 17 vs. 1, *p* < 0.001.
United Arab Emirates	El-Lababidi et al., 2017 [162]	An ASP was introduced in this single quaternary hospital; ASP included decision support systems, rapid diagnostics, and electronic surveillance systems; Overall antimicrobial consumption and costs monitored from 3rd quarter 2015 to 1st quarter 2017; Key areas included targeted antibiotics (typically Watch).	Carbapenem (**W**) use decreased by 32% from 105 to 71 DOT/1000 inpatient days (*p =* 0.05); Anti-MRSA agent prescribing decreased by 57% from 109 to 46 DOT/1000 inpatient days (*p =* 0.003); Anti-pseudomonal β-lactam prescribing decreased by 49% from 84 to 43 DOT/1000 inpatient days (*p =* 0.015); Total cost of antimicrobial therapy decreased by 67% from 323 AED/1000 inpatient days to 105 AED/1000 inpatient days (*p =* 0.01); Total cost total saving of 943,324 AED (USD 256,338) since program inception.
United Arab Emirates	El-Lababidi et al., 2019 [146]	ASPs introduced through a multidisciplinary team running from 1 July 215 to 31 December 2017; Key measures included decreasing the prescribing of antibiotics of concern and reducing costs.	4123 interventions were performed during the study period (with a 91% acceptance rate for recommendations; Usage and costs of monitored antimicrobials decreased despite a significant increase in patient discharges and total patient days. Hospital-onset *Clostridioides difficile* infection rates decreased from 0.46 cases/1000 patient days pre-ASP to 0.12 cases post-ASP (*p* = 0.035); Hospital-onset infections due to MDR organisms decreased from 2.39 cases/1000 patient days pre-ASP to 0.38 cases post-ASP (*p* = 0.05); Total direct cost savings estimated at USD 1,339,499 over the study period despite an increase in patient discharges and total patient-days; Antimicrobial costs/inpatient-day decreased by 32% from USD 47.2 pre-ASP to USD 32.3 post-ASP.
	Alshehhi et al., 2021 [67]	SAP guidelines were developed based on international guidelines Infectious Diseases Society of America, Society for Healthcare Epidemiology of America, American Society of Health System Pharmacists (ASHP) and the Centers for Disease Control and Prevention—facilitated by the clinical pharmacist; The clinical pharmacist also: ○ Followed the guideline’s approval process with the quality department; ○ Developed educational materials and conducted educational lectures during the study period, and disseminated these on the hospital intranet; ○ Sent official reminder emails to physicians to encourage adherence to the guidelines.	493 patients were included in the study; Antibiotics for SAP were typically administered for 3 days (104 patients; 32.9%), 5 days (89 patients; 25.7%), and 7 days (41 patients; 11.8%) prior to the ASP; Post-ASP antibiotics were administered for mainly 3 days (118; 80.8%) followed by 5 days (10; 6.8%), and 7 days (6; 4.1%); The average duration of cefuroxime (**W**—most prescribed antibiotic) pre-ASP was 3.8 ± 1.2 days—reducing to 2.6 ± 0.6 days post-ASP; Mean length of hospital stay was 5.31 ± 6.02 days pre-implementation falling to 4.90 ± 3.72 days post-ASP.
	Sadeq et al., 2021 [82]	For the intervention, patients who received antibiotics were reviewed by the ASP team consisting of ID physicians, ID clinical pharmacist and ward clinical pharmacists; Initially, the ward clinical pharmacists reviewed all adult patients prescribed antibiotics admitted to the medical, intensive care, or burns units for potential; ASP intervention: ○ The ward clinical pharmacist discussed a possible intervention with the ward physician or ID clinical pharmacist if needed; ○ Finally, the ID physician was approached for review and recommendations; ○ For some complicated cases, e.g., bacteremia, endocarditis, or CNS infections, ID physicians directly reviewed the cases and provided their recommendations.	3000 patients in total, pre- and post-intervention In the intervention group vs. the retrospective group there was: A reduction in the length of hospital stay (*p* < 0.01); A reduction in readmissions (*p* < 0.01); A reduction in mortality rates (*p* < 0.01); The use of “Reserve” group of antibiotics decreased in the intervention group (relative rate change = 0.88); The intravenous to oral antibiotic ratio in the medical ward decreased from 4.8 to 4.1.

NB: AMR: antimicrobial resistance; AMS: antimicrobial stewardship; ASP: antimicrobial stewardship program; AWaRe classification for antibiotics—Access, Watch, or Reserve [45]; ID = infectious disease; SAP: surgical antibiotic prophylaxis; SSIs: surgical site infections; * World Bank Status (Based on [131]).

**Table 5 antibiotics-12-00827-t005:** Knowledge, attitude, and perceptions among key stakeholders regarding antibiotics, AMR, and ASPs among countries in MENA.

Country	Author, Year	Study Design	Population	Objectives	Knowledge	Attitude/Perception	Practice	Inference
Jordan	Al-Qerem et al., 2022 [167]	Quantitative	Pharmacy students	AMR is a major health threat and efforts should be intensified to reduce its burden. Healthcare providers, especially pharmacists, can be actively involved in the reduction of AMR.	Appreciable number of students have knowledge of antibiotics (>60%).	Positive attitude	Overall good knowledge. However, concerns with practice in some areas, including the use of antibiotics and their disposal.	Universities should ensure that pharmacy students acquire adequate education about antibiotics, including their use.
Jordan	Al-Tani et al., 2022 [168]	Quantitative	Pharmacists	To assess the knowledge, opportunity, motivation, and behavior of pharmacists and their information.	Respondents score highly on effective use of antibiotics and side effects (87%). One-third reported no knowledge of any initiatives on antibiotics or AMR.	Positive attitude	Pharmacists indicated an interest in receiving more information on AMR and medical conditions where antibiotics are appropriate.	Require more knowledge of antibiotics, their appropriate use, and AMR.
Jordan	Al-Tani et al., 2022 [169]	Quantitative	Medicine, nursing, and Pharmacy students	Survey students’ knowledge, attitude, and practice regarding antimicrobial use and AMR.	Knowledge of more than three-quarters of respondents was good regarding antibiotics and side effects.	Positive attitude	Low awareness of the national action plan on AMR.	Require more information regarding antibiotic use and AMR.
Jordan	Karasneh et al., 2021 [170]	Quantitative	Physicians and dentists	Ascertain prescribers’ knowledge, attitudes, and behaviors about antibiotic use and AMR.	Prescribers had good knowledge of antibiotics, indications, and side effects (>90%). Lower knowledge of AMR (62.2%).	Positive attitude	Strategies not implemented effectively.	Educate practitioners and update them about local and global antibiotic use status and AMR.
Iran	Hashemzaei et al., 2021 [171]	Quantitative	Pharmacy and medical students	To investigate the knowledge, attitude, and practice of pharmacy and medical students toward self-medication.	There was no difference in the level of knowledge, mostly associated with years of study. Pharmacy students had better knowledge.	Pharmacy students had more negative attitudes than medical students.	Concerns with following national protocols.	The high prevalence of self-medication and the overuse of antibiotics can pose a significant risk of AMR.
Iran	Sami et al., 2022 [172]	Quantitative	Physicians	To assess KAP in Physicians.	97.2% were aware of the AMR problem in Iran.	95.6% agreed that prescribing of antimicrobials was not appropriate in Iran.	65.9% said before prescribing, they used local/ international guidelines <50% were in contact with a microbiology laboratory to guide prescribing.	Physicians’ level of knowledge about AMR and antimicrobial stewardship is poor. Consequently, a need to increase training on AMR and ASPs.
Lebanon	Zahreddine et al., 2018 [173]	Quantitative	Pharmacists and parents	To assess knowledge of both parents of children and pharmacist on antibiotics and AMR.	One-third of pharmacists did not know which factors were associated with AMR. Parents with university education had better knowledge of antibiotics.	Positive attitude	Misuse of antibiotics mostly involves parents, physicians, and pharmacists and not adhering to guidelines.	Implement educational campaigns in order to increase awareness of antibiotics, their misuse, and their implications on AMR.
Palestine	Al-Halawa et al., 2023 [29]	Quantitative	Community pharmacists	Evaluate the knowledge, attitude, and practices of community pharmacists to AMR.	Appreciable number (92.1%) said that inappropriate use increases resistance, and 86.2% disagreed that patients should stop antibiotics when symptoms improve.	Generally positive, with 75.6% agreeing that AMR is a serious public health issue.	Need to design programs to further improve the education of pharmacists and decrease dispensing of antibiotics without a prescription.	Re-design education and training, and strengthen legislation to reduce dispensing of antibiotics without a prescription.
Palestine	Jabbarin et al. [174]	Quantitative	Physicians	Evaluate the knowledge, attitude, awareness, and perceptions of AMR among physicians; and the correlation between their knowledge of AMR and experience.	Variable knowledge and perceptions of AMR. Senior specialists/consultants more knowledgeable about AMR.	Generally positive attitude to AMR, with 69.3% perceiving AMR as a very important problem worldwide and 54.7% a very important problem in the country.	A need to increase education on AMR among physicians with an emphasis on junior physicians Instigate activities to raise physicians’ awareness regarding AMR and its consequences on public health.	Implement educational programs among both practicing physicians and students to enhance knowledge of AMR and its public health importance.
Saudi Arabia	Akbar et al., 2021 [175]	Quantitative	Clinical laboratory science, nurses, and pharmacy students	Evaluate the knowledge of future HCWs in Saudi Arabia on antibiotics, antibiotic use, and antibiotic resistance.	Students have above-average knowledge of antibiotics and AMR However, misconceptions about antibiotics/ their use.	Happy to change.	Need improvements, especially in antibiotic use.	Curricular contents must be reviewed and enhanced to suit the specific learning needs of students.

**Table 6 antibiotics-12-00827-t006:** Suggested strategies to improve future antimicrobial utilization among countries in the MENA region.

*Short to Medium Term (1–5 Years)*
The MENA region’s governments, ministries, and health organizations must be dedicated to significantly reducing the inappropriate use of antibiotics in all healthcare facilities, including hospitals; Building the necessary infrastructure, including electronic records, information and clinical classification standards, business rules, data coding, and interoperability, as well as the resources (technical, personnel, and financial) to regularly collect prescribing data to monitor prescribing against agreed targets and for use in ASPs and NAPs; Ensuring governance of patient data by developing, monitoring, and enforcing privacy and security standards that stipulate administrative and technical rules to protect sensitive health data from misuse, unauthorized access, or disclosure; Regularly monitor prescribing versus agreed-upon prescribing/quality parameters, building on the introduction of additional ASPs, and then feedback the findings. Targets should be increasingly based on recognized guidance such as the AWaRe book [48,51]; Prescribing and quality indicators for use in various hospitals across a nation should be agreed upon with all major stakeholder groups, drawing on the fundamental tenets of indicator development. Starting points include currently used evidence-based prescribing/quality indicators. To avoid overload, it is necessary to make sure that any agreed-upon quality indicators are kept to a minimum and subsequently given priority over other activity measures. The type of existing and future patient record keeping, including electronic healthcare systems, and how frequently the data are collected/prescriptions should be monitored to determine the content and nature of any agreed quality indicator; Any agreed-upon metrics must be integrated into ongoing ASPs in hospitals. To increase the likelihood of ASPs succeeding, the necessary training, financial resources, and manpower must be made available, including recruiting and training digital data analysts. This should be part of NAPs with planned and ongoing ASP activities pertaining to the WHO AWaRE book and guidance—especially if there are issues with present limited activities inside hospitals and a lack of knowledge and expertise within hospitals regarding the AWaRe classification and book as well as ASPs; As part of this program, enhance the role of hospital Drug and Therapeutic Committees (DTCs) alongside AMS teams throughout the MENA region. Functioning AMS teams and DTCs are vital going forward where there have only been sporadic operations to date—this may entail increasing resources and training where pertinent; Key target areas for ASPs in hospitals include addressing the use of prolonged prophylaxis to prevent SSIs as well as the general overuse of antibiotics in patients admitted to hospital with COVID-19/other acute respiratory infections. This also includes enhancing the prescribing of “Access” vs. “Watch” and “Reserve” antibiotics [48,51]; Monitoring in accordance with current standards and NAPs, strengthened by academic detailing, auditing, and the use of computerized record-keeping systems. Moreover, organizations, including the Commonwealth Pharmacists, should create and test tools to help with prescribing and ASPs; Increasing the training of medical, dental care, and other allied healthcare students regarding antibiotics, AMR, and ASPs, where this is currently sub-optimal. Post qualification, make sure doctors, hospital pharmacists, microbiologists, and other critical medical personnel are knowledgeable about antibiotics, AMR, and ASPs. This is increasingly likely to incorporate hybrid learning that builds on the lessons learned from the COVID-19 epidemic.
** *Long Term (5–10 Years)* **
As part of agreed-upon NAPs across the MENA region, routinely track trends in antimicrobial use and resistance patterns across sectors; Where appropriate, implement additional multiple strategies to improve the appropriate prescribing of antibiotics in hospitals, including the establishment of IPC/ASP committees, routine CST, the creation and regular updating of hospital-specific antibiograms following internationally recognized guidelines, the implementation of clinical decision support systems as part of ASPs and ongoing updating of recommendations; Identify and monitor the use of agreed digital codes for diagnoses linked to the AWaRe classification and book to improve future prescribing; Creating new quality indicators or improving existing ones, as necessary, while avoiding overloading healthcare professionals and digital analysts; Countries in MENA require new antimicrobials as well as keeping resistance down regarding current antimicrobials, proactively addressing shortages/stockouts as well as making sure the necessary diagnostic equipment is routinely available; Review curricula and other educational initiatives at medical, pharmacy, nursing, and laboratory training schools on a regular basis to assess students’ knowledge of antibiotics, ASPs, and AMR, and instigate changes to the curricula where needed; Hospital quality improvement initiatives include requiring prescribers to regularly justify their treatment choices, increasing prescriber accountability, increasing dialogue between laboratories and prescribers, and, when appropriate, placing restrictions on the use of specific antibiotics in accordance with the WHO AWaRe list and agreed-upon quality indicators; While awaiting culture and sensitivity testing results, continue to create, update, and disseminate hospital antibiograms to optimize empiric prescribing; Continue collaborating with important stakeholders to increase adherence to established national and local recommendations, reporting to external bodies such as the WHO GLASS surveillance system, in order to enhance patient outcomes and lower AMR.

NB: AMS: antimicrobial stewardship; ASP: antimicrobial stewardship program; CST: Culture Sensitivity Test; IPC: infection, prevention, and control; NAP: national action plan; SAP: surgical antibiotic prophylaxis; SSIs: surgical site infections.

## Data Availability

Additional data is available upon reasonable request from the corresponding author. However, all informational sources and papers have been extensively referenced.

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
