# Peer review of "Ongoing Strategies to Improve Antimicrobial Utilization in Hospitals across the Middle East and North Africa (MENA): Findings and Implications"

_antibiotics, 2023, doi:10.3390/antibiotics12050827_

Round 1

Reviewer 1 Report

In this manuscript the authors have compiled the ongoing as well as the previously published studies with respect to improved antibiotic utilization in an effort to reduce antimicrobial resistance, especially in the Middle eastern and North African countries.

Line 43: Slow down instead of "slow"

Line 159-160: Please rephrase, sentence feels incomplete.

Line 176-179: It is unclear if the mentioned hospitals were excluded from the analysis.

The studies described with respect to ASP are very well described. 

Author Response

Comments and Suggestions for Authors

1) In this manuscript the authors have compiled the ongoing as well as the previously published studies with respect to improved antibiotic utilization in an effort to reduce antimicrobial resistance, especially in the Middle eastern and North African countries.

Author comments: Thank you for this summary – appreciated!

2) Line 43: Slow down instead of "slow"

Author comments: Thank you – now changed

3) Line 159-160: Please rephrase, sentence feels incomplete.

Author comments: Thank you – now updated. We hope this is now OK.

4) Line 176-179: It is unclear if the mentioned hospitals were excluded from the analysis.

Author comments: Thank you – now made clearer

5) The studies described with respect to ASP are very well described. 

Author comments: Thank you – appreciated!

Reviewer 2 Report

A brief summary 

The manuscript presents a very important topic of great importance in a geographical region. They examined current hospital antimicrobial consumption trends  across the Middle East along with documented ASPs.

General concept comments

1.     Is the manuscript clear, relevant for the field and presented in a well-structured manner?

yes

  1. Are the cited references current (mostly within the last 5 years)?

I didn't have time to check, there are 200 (!!!) references in the article. Based on an estimate, most of them have recent (within the last 5 years) references.

3.     Does it include an abnormal number of self-citations?

I don't know, there are a total of 21 self-citations, which is 10.5%, I think this is more than usual.

4.     Is the manuscript scientifically sound and is the experimental design appropriate to test the hypothesis?

yes

5.     Are the manuscript’s results reproducible based on the details given in the methods section?

Yes.

6.     Are the figures/tables/images/schemes appropriate? Do they properly show the data? Are they easy to interpret and understand? Are the data interpreted appropriately and consistently throughout the manuscript? Please include details regarding the statistical analysis or data acquired from specific databases.

Yes,

7.     Are the conclusions consistent with the evidence and arguments presented?

Yyes.

8.     Please evaluate the ethics statements and data availability statements to ensure they are adequate.

 Yes, it is adequate.

Author Response

Comments and Suggestions for Authors

A brief summary 

The manuscript presents a very important topic of great importance in a geographical region. They examined current hospital antimicrobial consumption trends across the Middle East along with documented ASPs.

Author comments: Thank you for your kind words – appreciated

General concept comments

1) Is the manuscript clear, relevant for the field and presented in a well-structured manner? - Yes

Author comments: Thank you

2) Are the cited references current (mostly within the last 5 years)? - I didn't have time to check, there are 200 (!!!) references in the article. Based on an estimate, most of them have recent (within the last 5 years) references.

Author comments: Thank you

  1. Does it include an abnormal number of self-citations? I don't know, there are a total of 21 self-citations, which is 10.5%, I think this is more than usual.

Author comments: Thank you for this comment. We have been through the list of publications and deleted a number of self-citations that we believed did not add relevance to the paper. These have now been replaced by recent papers (not by the co-authors) that add further depth to the paper. We hope this is now acceptable.

  1. Is the manuscript scientifically sound and is the experimental design appropriate to test the hypothesis? - yes

Author comments: Thank you

  1. Are the manuscript’s results reproducible based on the details given in the methods section? - Yes.

Author comments: Thank you

  1. Are the figures/tables/images/schemes appropriate? Do they properly show the data? Are they easy to interpret and understand? Are the data interpreted appropriately and consistently throughout the manuscript? Please include details regarding the statistical analysis or data acquired from specific databases. – Yes

Author comments: Thank you

  1. Are the conclusions consistent with the evidence and arguments presented? yes.

Author comments: Thank you

  1. Please evaluate the ethics statements and data availability statements to ensure they are adequate. Yes, it is adequate.

Author comments: Thank you

Reviewer 3 Report

In this narrative review, the authors bring to attention the threat of worsening antibiotic resistance with continued inappropriate antibiotic use in the MENA region and the crucial role of stewardship interventions to counteract this.

Please note some suggestions regarding the manuscript below

- in the results section it is mentioned that most commonly used antibiotics are 3rd gen cephalosporins, penicillins and vancomycin (lines 190-191). In the discussion section however, metronidazole is mentioned as the 3rd most commonly used antibiotic and not vancomycin (lines 311-312). Are the authors referring to a different study, please clarify. 

- unclear what Table 3 adds to the review. It mainly talks about SSI risk factors and causative pathogens. May be it can be moved to the supplementary section. 

- please ensure information in the tables are accurate. For example, in table 6, reference 29 (Al-Halawa et.al 2023) it says that 86.2% reported that patients should stop antibiotics when symptoms improve but the study abstract reports the opposite - that 86.2% disagreed that patients should stop antibiotics when symptoms improve. 

Author Response

Comments and Suggestions for Authors

1) In this narrative review, the authors bring to attention the threat of worsening antibiotic resistance with continued inappropriate antibiotic use in the MENA region and the crucial role of stewardship interventions to counteract this.

Author comments: Thank you for these positive comments – appreciated!

Please note some suggestions regarding the manuscript below

2) In the results section it is mentioned that most commonly used antibiotics are 3rd gen cephalosporins, penicillins and vancomycin (lines 190-191). In the discussion section however, metronidazole is mentioned as the 3rd most commonly used antibiotic and not vancomycin (lines 311-312). Are the authors referring to a different study, please clarify. 

Author comments: Thank you. This was typing mistake. We have now made the changes. We hope this is now acceptable.

3) Unclear what Table 3 adds to the review. It mainly talks about SSI risk factors and causative pathogens. May be it can be moved to the supplementary section. 

Author comments. Thank you for this. We have now moved this Table to Supplementary Material (for back-up for readers interested in this). We hope this is now OK.

4) Please ensure information in the tables are accurate. For example, in table 6, reference 29 (Al-Halawa et.al 2023) it says that 86.2% reported that patients should stop antibiotics when symptoms improve but the study abstract reports the opposite - that 86.2% disagreed that patients should stop antibiotics when symptoms improve. 

Author comments: Thank you. This was typing mistake. We have made the changes and highlighted this. In addition, we have checked the other references in the Tables and clarified areas where concerns. We hope this is now OK.
